# Mechanisms of Cadmium Neurotoxicity

**DOI:** 10.3390/ijms242316558

**Published:** 2023-11-21

**Authors:** Madelyn A. Arruebarrena, Calvin T. Hawe, Young Min Lee, Rachel C. Branco

**Affiliations:** 1Neuroscience and Behavior Program, University of Notre Dame, Notre Dame, IN 46556, USA; marrueba@nd.edu (M.A.A.); ylee26@nd.edu (Y.M.L.); 2Department of Chemistry and Biochemistry, University of Notre Dame, Notre Dame, IN 46556, USA; chawe@nd.edu

**Keywords:** metals, enzyme function, glycogen, neurodegeneration, neurotoxicity, cadmium, neurotransmission

## Abstract

Cadmium is a heavy metal that increasingly contaminates food and drink products. Once ingested, cadmium exerts toxic effects that pose a significant threat to human health. The nervous system is particularly vulnerable to prolonged, low-dose cadmium exposure. This review article provides an overview of cadmium’s primary mechanisms of neurotoxicity. Cadmium gains entry into the nervous system via zinc and calcium transporters, altering the homeostasis for these metal ions. Once within the nervous system, cadmium disrupts mitochondrial respiration by decreasing ATP synthesis and increasing the production of reactive oxygen species. Cadmium also impairs normal neurotransmission by increasing neurotransmitter release asynchronicity and disrupting neurotransmitter signaling proteins. Cadmium furthermore impairs the blood–brain barrier and alters the regulation of glycogen metabolism. Together, these mechanisms represent multiple sites of biochemical perturbation that result in cumulative nervous system damage which can increase the risk for neurological and neurodegenerative disorders. Understanding the way by which cadmium exerts its effects is critical for developing effective treatment and prevention strategies against cadmium-induced neurotoxic insult.

## 1. Introduction

Cadmium is a highly toxic pollutant that permeates environmental, industrial, and agricultural spaces. The Agency for Toxic Substances and Disease Registry ranked cadmium as the seventh most hazardous substance to human health [1], and the Department of Health and Human Services listed cadmium as a known human carcinogen in 2021 [2]. Recent anthropogenic activities have increased human exposure to cadmium. Most commercial cadmium is a byproduct of zinc ore mining that is used in electroplating, battery production, paint pigments, and plastics [3,4]. These activities introduce cadmium to the agricultural sphere, where plants readily absorb cadmium from contaminated soil and water. Additionally, cadmium contamination of ethanol is common, with variable levels detected in wine, beer, whiskey, gin, and other alcoholic products [5]. As a result, the most common source of exposure for the general population is contaminated food and drink products [3].

Cadmium enters the human body by various routes. Uptake is facilitated by the ingestion of contaminated food and beverage products, the inhalation of aerosolized cadmium particles in cigarette smoke, and particle accumulation in the olfactory bulb following industrial fume exposure [6,7]. Due to its abiogenic nature, cadmium has no endogenous mechanism of clearance and thus exhibits a low urinary excretion rate. It accumulates in the human body with an estimated half-life of up to 23.5 years [8]. As a result of this accumulation, the estimated mass of cadmium within adults in the U.S. and Europe who have not been occupationally exposed to cadmium is between 9.5 mg and 40 mg [9]. Moreover, blood concentrations of cadmium were found to be ~0.4 µg/L [10] and cerebrospinal fluid (CSF) concentrations of cadmium were found to be 72 ng/L in humans [10,11]. Thus, CSF concentrations of cadmium in humans are only roughly five-fold lower than in blood.

Chronic accumulation of cadmium results in multiorgan toxicity, primarily targeting the kidney, skeleton, liver, and nervous system [12], reviewed in [9]. Among these, the nervous system is a particularly vulnerable target for cadmium toxicity. Cadmium can increase risk of peripheral neuropathy, altered equilibrium, and poor performance on visuomotor tasks [13]. Exposure to cadmium is correlated with reduced concentration, poorer cognitive function in older adults, and adverse learning outcomes in children [13,14,15,16]. Cadmium exposure has also been associated with neurodegenerative disease pathologies observed in Alzheimer’s disease (AD), Parkinson’s disease (PD), and amyotrophic lateral sclerosis (ALS) [17,18,19]. Cadmium exerts its neurotoxic outcomes via diverse means [12,20,21] (Figure 1). Here, we review the current knowledge concerning the sites of exogenous cadmium insult that result in nervous system dysfunction.

## 2. Cadmium Entry to the Nervous System

Cadmium gains entry to the nervous system primarily by oral ingestion, at which point it is absorbed into the bloodstream and can damage the blood–brain barrier (BBB) to accumulate within nervous system tissue. Cadmium inhalation provides an even more direct route to the nervous system, since the olfactory epithelium lacks protection offered by the BBB and permits cadmium uptake directly into nervous tissue [22]. Cadmium is similar to bioessential metal cations implicated in neuronal transmission, particularly calcium and zinc. Cadmium, calcium, and zinc are primarily divalent cations that possess similar chemical properties and favor the oxidation state of +2. Calcium and cadmium share similar ionic radii (0.97 Å and 0.99 Å, respectively) and charge/radius ratios (Ca^2+^ = 2.02 e/Å, Cd^2+^ = 2.06 e/Å), granting each the ability to exert similarly strong electrostatic forces on biogenic macromolecules (reviewed in [23]). Cadmium and zinc are elements in Group IIB of the periodic table with the same electron configuration, allowing similar chemical behavior within ion-protein interaction. In this way, cadmium can permeate nervous system cells and organelles by taking advantage of endogenous zinc- and calcium-specific transporters.

Several studies have implicated cadmium as a competitive voltage-gated calcium channel (VGCC) inhibitor [24,25,26]. Cadmium enters the rat cerebellar granular neuron primarily through dihydropyridine-sensitive (L-type) VGCCs as it competes with Ca^2+^ for within the channel pore. Exposure to 100 µM cadmium prevented an increase in cytosolic calcium concentration after neuronal depolarization, and cadmium was able to permeate the neuron. The N-type VGCC is also implicated in cadmium-induced blockage of Ca^2+^ current in frog sympathetic neurons [27]. Cadmium completely and rapidly blocked Ca^2+^ current at voltages when Ca^2+^ channels are primarily open (0 to +30 mV), indicating that the N-type VGCC is a route of cadmium entry into sympathetic neurons. Because VGCCs are densely concentrated at the presynaptic site, the presynaptic terminal is a notable location of cadmium uptake in neuronal cells (reviewed in [28]).

Cadmium also enters neuronal cells through zinc transporters, the most significant of which are the ZIP6 and ZnT3 transporters [28,29]. ZIP6, an importer, is localized to the plasma membrane of hippocampal pyramidal neurons while ZnT3 is an exporter plentiful on the presynaptic neuronal membrane that regulates the brain’s vesicular pool [30,31], reviewed in [22]. Mimouna et al. found that early-life cadmium exposure increased cadmium accumulation in the brain, increased ZIP6 gene expression, and decreased ZnT3 expression [29]. The simultaneous upregulation of the ZIP6 importer and downregulation of the ZnT3 exporter may lead to cadmium accumulation in these neurons. In a later study, Mimouna et al. investigated interactions between cadmium and ZnT3 in hippocampal neurons. Treatment of rat hippocampal neurons with cadmium chloride (0, 0.5, 5, 10, 25, or 50 µM) and zinc chloride (0, 10, 30, 50, 70, or 90 µM) for either 24 or 48 h downregulated ZnT3 mRNA expression, an effect attenuated by the application of zinc. Zinc supplementation at 30 µM significantly ameliorated cadmium-induced neurotoxicity in cells treated with 10 and 25 µM cadmium [32]. Presumably, the physicochemical similarities between cadmium and zinc allow cadmium to enter synaptic vesicles through ZnT3 and accumulate, ultimately resulting in cell death and disruption of neuronal plasticity. 

## 3. Cadmium Effects on Mitochondrial Respiration 

Mitochondria in the nervous system perform critical roles not only in energy production [33] but also in neuronal development, function, and survival [34]. Neurons, the functional unit of the nervous system, are particularly high consumers of ATP due to their constant need to maintain the neuronal concentration gradient necessary for action potential propagation, operate the cellular machinery associated with the vesicle cycle, facilitate axonal transport, and provide energy for synaptic plasticity [33,34,35]. Thus, any disruption in mitochondrial function can result in energy deficits, significantly compromising neural activity and health.

Oxidative phosphorylation relies on a strong mitochondrial membrane potential (ΔΨm) in order to produce ATP via ATP synthase [35]. The electron transport chain (ETC), embedded within the inner mitochondrial matrix uses the potential energy from electron-carrying molecules in order to produce a robust ΔΨm. The four protein complexes that comprise the ETC must deftly handle redox molecules in order to appropriately produce a proton gradient, the basis of the ΔΨm. Furthermore, reactive oxygen species (ROS) are produced at low concentrations as a byproduct of the ETC. Low levels of ROS can be mitigated by antioxidant molecules within the mitochondria, such as glutathione. However, if ROS are allowed to proliferate, either via external influence or inappropriate regulation of the ETC, the resultant oxidative stress results in cellular damage.

This ΔΨm gradient can be regulated via mitochondrial uncoupling proteins, which can serve to respond to cellular energetic needs, maintain consistent temperature, or control osmotic swelling. However, various pathological conditions can disrupt ΔΨm, leading to impaired mitochondrial respiration. For instance, mitochondrial permeability transition pore (PTP) opening can be triggered by factors like oxidative stress that can result in ΔΨm depolarization [36]. Such depolarization can inhibit ATP synthesis and compromise overall mitochondrial function.

The significance of mitochondria becomes most apparent in the context of neurodegenerative diseases, including AD, PD, and ALS. These conditions are characterized by mitochondrial dysfunction [35]. Abnormalities encompass impaired energy production, heightened ROS production, and compromised calcium handling, collectively contributing to neuronal degeneration and the clinical manifestations of these diseases [36]. 

### 3.1. Cadmium Interference with the Electron Transport Chain

The mitochondria have emerged as primary targets in cadmium toxicity (for an excellent review focusing exclusively on this topic, see [37]). This is supported by experimental evidence in a rodent model, where cadmium exposure on isolated mitochondria from mouse livers led to extensive organelle damage [38]. One mechanism by which cadmium disrupts mitochondrial function is by interfering with specific protein complexes within the ETC such that ΔΨm is reduced and the proton-motive force that drives ATP synthesis is subsequently weakened. Cadmium interacts with Complex I of the ETC at both the Q-binding site and the NADH-binding site, decreasing the ability of Complex I to shuttle electrons and transport protons to create and maintain ΔΨm. Cadmium’s interaction with the Q_o_ site of Complex III redirects ROS production toward the intermembrane space, effectively bypassing the matrix antioxidant defenses [39,40]. By disrupting the normal function in Complexes I and III, the resultant decrease in ΔΨm ultimately leads to a decreased ability to efficiently synthesize ATP and increase in damaging cytosolic ROS. 

### 3.2. Cadmium Opens the Permeability Transition Pore

Furthermore, cadmium induces the opening of the permeability transition pore (PTP), a dynamic protein complex residing at the interface between the inner and outer mitochondrial compartments [41]. The PTP allows for the diffusion of small molecules through the inner mitochondrial membrane, dissipating the ΔΨm and thereby halting ATP synthesis. Opening of the PTP also acts as a signal for apoptosis via release of stores of cytochrome C. The weakening of the inner mitochondrial gradient itself can trigger opening of the PTP in a feedforward mechanism that results in eventual cell death. It is not clear to what extent cadmium opens the PTP via weakening of the ΔΨm in mechanisms described above, or whether cadmium directly interacts with the PTP itself to increase the likelihood of opening, or both. 

There is some evidence to suggest that cadmium directly interacts with the PTP to increase opening, independent of cadmium’s effects on ΔΨm. Cadmium interacts with a constituent of the PTP complex, the adenine nucleotide translocator (ANT), at the thiol groups present on the cysteine residues, potentially leading to modifications of ANT function [39]. ANT exchanges cytosolic ADP and matrix ATP, enabling cytosolic ATP export out of the mitochondria while delivering ADP to the mitochondria [39,40]. Structural studies have shown that ADP/ATP exchange of ANT proteins occurs via an “induced transition fit” model. This process begins with ADP binding at the “c-state”, where the protein is exclusively open to the intermembrane space. This binding triggers a conformational shift to “m-state”, where the protein becomes exclusively open to the mitochondrial matrix, facilitating the exchange of ADP for ATP [42]. Inhibiting ANT blocks this cadmium-induced PTP opening [43]. This cadmium-induced PTP opening can also be blocked via addition of an inhibitor of the mitochondrial calcium importer, indicating that cadmium is gaining access to the inner mitochondria via this transporter [39].

Furthermore, while calcium can induce PTP opening via a cyclosporin A (CsA)-dependent mechanism [43], cadmium opens the PTP independent from this calcium-CsA pathway. Because increased calcium is a potent intracellular signal within neurons for apoptosis, cadmium’s ability to bypass calcium-induced mechanisms of apoptosis represent an alternative pathway for unregulated neuronal death. The regulatory mechanisms of this CsA-independent apoptotic pathway are as yet unclear. 

### 3.3. Recent Focuses of Mitochondrial Apoptosis/Dysfunction: ER Stress and SIRT1 

It is increasingly clear that the relationship between mitochondrial stress and endoplasmic reticulum (ER) stress both contribute to cadmium’s toxic effects. According to multiple studies, cadmium exposure may induce crosstalk between the stress responses of the ER and mitochondria, which culminates in cell apoptosis [44,45]. One important aspect of this crosstalk is the involvement of proapoptotic proteins Bim and Bax, which are upregulated following acute cadmium exposure. Bax, in particular, translocates from the cytosol to the mitochondria causing the apoptosis of the mitochondria. Furthermore, cadmium exposure leads to the release of cytochrome c from the mitochondria to the cytoplasm. This release triggers apoptotic signaling and activates caspases, leading to cell apoptosis [44,46,47].

Furthermore, exposure of cadmium to human cell lines led to an increase of intracellular ROS levels in a dose dependent manner. This generation of ROS occurred in a feed-forward fashion that ultimately induces GADD153, a marker that initiates cell death [45]. A protective measure against this process is the antioxidant resveratrol, which inhibits ER stress and GADD153 and activates sirtuin1 (SIRT1) [48]. 

More recent research has also pointed towards SIRT1 as a critical regulator of the biochemical response to oxidative stress [45]. SIRT1 is a nicotinamide dinucleotide (NAD^+^)-dependent deacylases known for its ability to regulate cellular processes such as DNA repair, inflammation, fatty acid oxidation, fat differentiation, and more [48]. This suppression of SIRT1 by cadmium leads to a marked increase in oxidative stress within neuronal cells. The ensuing oxidative stress disrupts mitochondrial function, which, in turn, culminates in the death of neural cells. This phenomenon has been observed in both PC12 cells, a neuron-like cell line, and primary rat cerebral cortical neurons [45]. Activating SIRT1 prevented the buildup of ROS and cellular loss and expounds on a potential mechanism by which SIRT activators affect SIRT1 activity, particularly by deacetylating PGC-1a. This deacetylation is believed to contribute to the enhancement of oxidative metabolism, playing a crucial role in the cellular response to oxidative stress [49]. SIRT1 is structurally important for the nervous system as it promotes axonal elongation, neurite outgrowth, and dendritic branching. Furthermore, it has been found to be crucial for memory formation and its protective measures against neurodegenerative diseases such as Alzheimer’s, Parkinson’s, and motor neuron diseases [50,51,52,53]. 

The precise molecular mechanism underlying cadmium-induced neurotoxicity in the context of mitochondria-associated ER membranes (MAMs) remains unclear. MAMs consist of a diverse array of proteins, including mitofusin 2 (Mfn2), voltage-dependent anion channel (VDAC), and glucose-regulated protein 75 (Grp75). These proteins facilitate the transport of calcium ions from the ER to the mitochondria through the inositol 1,4,5-triphosphate receptors (IP3R) on the ER and the voltage-dependent anion-selective channel protein (VDAC) on the mitochondria [49]. Exposure to cadmium increased expression of Mfn2, Grp75, and VDAC1 [52]. Additionally, both PC12 cells and primary neurons exhibited a significant reduction in mitochondrial calcium uptake when Mfn2 was knocked out in response to cadmium treatment. Notably, this unveiled that the principal driver of cadmium-induced autophagy in neuronal cells may be the uptake of mitochondrial calcium facilitated by MAMs, specifically that the IP3R-Grp75-VDAC1 complex is regulated by Mfn2. The interplay between Mfn2 and the operation of the IP3R-Grp75-VDAC1 complex represents a breakthrough in the understanding of mitochondrial dysfunction following cadmium exposure [50]. 

### 3.4. Cadmium-Induced Autophagy

Autophagy, a regulated form of cell death, involves a series of steps directing targeted materials to the lysosome for recycling (for an extensive review focusing on autophagy in neurodegenerative diseases, see [51]). Cadmium-induced autophagy is associated with neurodegenerative disease, though the nature of the relationship remains somewhat controversial [52,53,54]. Hence, this section focuses on recently published papers on the topic of cadmium-induced autophagy.

Since autophagy is a key process for eliminating excess protein, it is thought that disruption in autophagy process via cadmium can result in excess misfolded protein leading to neurodegenerative disease [12]. Cadmium triggers neuronal apoptosis through an increase in autophagosome formation, marked by elevated LC3-II and p62 in neuronal cells, resulting in neuronal apoptosis [55]. The drug rapamycin prevents cadmium-induced increase in LC3-II and p62. Cadmium-induced apoptosis is dependent on the overproduction of autophagosomes by preventing autophagosome–lysosome fusion [55,56,57]. However, other recent studies have shown that cadmium inhibits autophagy through calcium-dependent activation of the JNK signaling pathway in a cell culture model [58].

Recent research has advanced the study of ameliorative strategies for preventing cadmium-induced changes to autophagic flux. *Potentilla anserine*, an herb native to the Qinghai–Tibet Plateau of China, is renowned for its nutrient richness and application in Chinese medicine. Emerging research highlights *Potentilla anserine* polysaccharide (PAP), a major bioactive component of this herb, as a candidate to prevent oxidative stress, mitochondrial cell death, and apoptosis [59,60,61,62]. PAP potentially mitigates cadmium-induced neuronal death via autophagy by suppressing the PI3K class III/Beclin-1 signaling pathway [63]. Interestingly, drugs that increase autophagy also seem to have some promise in preventing cadmium-induced neurotoxic damage. Linagliptin, an FDA-approved antidiabetic drug used to treat type 2 diabetes, also shows neuroprotective effects against cognitive decline [64,65]. Studies of linagliptin’s neuroprotective effects against cadmium exposure in rats have shown that linagliptin prevented the cognitive deficit induced by cadmium. However, linagliptin stimulated the hippocampal AMPK/mTOR pathway, which positively impacts autophagy progression. It is thought that this increase in autophagy stimulated clearance of neuronal misfolded proteins, resulting in improvement in cognitive impairment in this context [66]. Together, these results point toward the need for the further exploration of cadmium’s role in autophagic processes.

## 4. The Role of Cadmium in Synaptic Transmission

The synapse itself is a vulnerable target for cadmium toxicity. For the efficient transmission of a neuronal signal, biological metal cations must act in conjunction with a series of voltage-gated and ligand-gated channels. Cadmium’s physicochemical similarities to these ions, particularly calcium and zinc, permit its neurotoxicity at the synaptic level as cadmium permeates the presynaptic neuron, induces oxidative stress, and ultimately aggravates neuronal degeneration. 

### 4.1. Cadmium-Induced Asynchronous Neurotransmitter Release

Synchrony of neurotransmitter release is a marker of efficacious neural communication. The release of a neurotransmitter occurs within hundreds of milliseconds following the action potential to ensure precise communication between neurons [28]. Indeed, several studies have linked asynchronous release to neurodegenerative disease pathologies in AD, spinal muscular atrophy (SMA), and ALS [67,68,69,70]. Cadmium may augment asynchronous neurotransmitter release, further aggravating these neurodegenerative disease pathologies.

Cadmium application of 0.1 µM desynchronized neurotransmitter release in the distal compartment of the frog nerve terminal. This asynchrony was accompanied by a sharp increase in mitochondrial ROS production and lipid peroxidation, suggesting that cadmium-induced oxidative stress co-occurs with this desynchronization. Desynchronization was completely blocked by the administration of antioxidants and NADPH-oxidase inhibitors [55]. One possible mechanism of this asynchrony relies on cadmium’s action as a VGCC antagonist in addition to its role as initiator of oxidative stress. Extracellular cadmium likely replaced native calcium as the metal ion flowing through L-type VGCCs. This decrease of calcium inward current in the presence of cadmium leads to a blunted presynaptic spike of cytosolic calcium, which is integral for the coordination of vesicular machinery. Therefore, in the presence of cadmium, VGCCs must remain open for a longer period of time to allow sufficient calcium influx for the initiation of calcium-dependent presynaptic processes. The prolonged period in which VGCCs are open may lengthen the delay observed between the arrival of the depolarizing action potential and neurotransmitter release, accounting for the observed asynchrony. 

### 4.2. Cadmium Disruption of Neurotransmission

In addition to delaying neurotransmitter release, cadmium disrupts neurotransmitter packaging within synaptic vesicles, decreasing the amount of neurotransmitter available for each release event. There is particular evidence for this in glutamatergic neurons. Vesicular transporters rely on the proton electrochemical gradient generated by V-ATPase to package neurotransmitters into vesicles [71,72]. A volume of 50 µM of cadmium in isolated Wistar rat synaptosomes caused the dissipation of the proton gradient necessary to package glutamate into its synaptic vesicles, resulting in decreased depolarization-evoked exocytosis of glutamate and reduced extracellular glutamate concentration [73]. Although the mechanism by which the V-ATPase is disrupted was not directly observed, interaction with the thiol groups in the cysteine residues of the V-ATPase is a likely culprit. 

Additionally, cadmium exposure has been observed to induce changes in cholinergic muscarinic receptors and acetylcholinesterase (AChE) variants [74]. Specifically, Cd^2+^ exposure documented an elevation of the gene expression of AChE-S (the synaptic variant) while reducing the gene expression of AChE-R (the readthrough variant). This modification in AChE variants has been linked to cell death in these neurons. Moreover, cadmium treatment disrupts muscarinic receptors, particularly the M1 and M3 receptors, which play crucial roles in the regulation of memory and learning processes. This interference with the receptors may contribute to the cognitive impairments observed following exposure to cadmium. Although the precise mechanisms by which cadmium alters muscarinic receptors and AChE variants remain incompletely elucidated, oxidative stress has been posited as a potential intermediary factor in this process.

## 5. Cadmium and Other Metals

### 5.1. Cadmium Disruption of Zinc Signaling and Homeostasis 

Zinc, which itself can protect against cadmium-induced hippocampal neurotoxicity [25], decreased quantal release and markedly desynchronized neurotransmitter release at a concentration of 25 µM. Zinc can function as either a prooxidant or antioxidant in cellular systems, with both excesses and deficiencies resulting in oxidative stress (reviewed in [75]). Zinc-induced oxidative stress has been connected to neurodegeneration and cell death in cultured cortical neurons [44,47] and AD [45]. The influx of cadmium through zinc transporters may disrupt this zinc allostasis, resulting in exacerbated oxidative stress. Therefore, zinc and cadmium may act synergistically to induce oxidative stress in presynaptic terminals, ultimately resulting in decreased quantal release and asynchrony that advance neurodegeneration. 

The downregulation of the zinc transporter ZnT3 resulting from cadmium exposure results in downstream effects that affect critical signaling pathways in the brain. This downregulation initiates a cascade that decreases hippocampal brain-derived neurotrophic factor-tropomyosin receptor kinase B (BDNF-TrkB) and Erk1/2 signaling, intracellular messengers that play integral roles in neuronal plasticity and growth [50]. The TrkB neurotrophin receptor and subsequent BDNF activation are essential for advancing neuronal plasticity, and antidepressant binding to neurotrophin receptors, particularly TrkB, has been previously evidenced to facilitate BDNF activation and initiate neuronal plasticity [51]. While antidepressants bound to the TrkB neurotrophin receptor aid neuronal plasticity via BDNF activation, other xenobiotics like cadmium may inhibit it via indirect mechanisms such as ZnT3 downregulation. Further research is necessary to elucidate the mechanisms by which cadmium impedes neuronal plasticity.

Neuronal senescence is a hallmark of cumulative cellular damage. However, the mechanisms of neuronal senescence are varied and complex. Oxidative stress and neuronal senescence have been closely linked in several studies [48,58,59]. The presence of excess ROS results in proteolysis that impacts cell function and manifests as aging. Garfinkel introduced the “zinc hypothesis of aging” [76]. According to this hypothesis, dietary zinc deficiency results in less zinc availability for its metalloenzymes, leading to metalloenzyme dysregulation. This dysregulation, which varies by cell type, precipitates protein malformation and accumulation, which ultimately manifests as aging. He later predicted that zinc deficiency was secondary to cadmium toxicity [77]. According to this hypothesis, cadmium may drive zinc dysfunction and ultimately catalyze neuronal senescence in both control and neurodegenerative models. 

In 2020, Xie et al. reviewed zinc’s role in the development of Alzheimer’s, explaining that the disruption of zinc homeostasis may have implications for AD. In the CNS, ZnT3 packages zinc into presynaptic vesicles of zincergic neurons concentrated in the hippocampus, amygdala, and cerebral cortex. Presynaptic release of zinc from zincergic neurons has been postulated to modulate neuronal plasticity and learning and memory [78]. However, post-mortem brain tissue analysis of AD patients revealed decreases in the mRNA and protein levels of ZnT3 [79,80]. ZnT3 downregulation prevents packaging of zinc into vesicles, resulting in an excess of intracellular zinc within the neuron that readily binds to amyloid-beta (Aβ) oligomers associated with AD pathogenesis. The binding of zinc to Aβ alters the secondary structure of Aβ such that it promotes the formation of neurotoxic spherical species [60] while also limiting the bioavailability of zinc for its role as plasticity modulator. AD patients exhibit zinc deficiency in serum, which may be explained by this continual interaction between zinc and Aβ that promotes neurotoxic oligomer and fibril formation, effectively sequestering zinc from fulfilling its natural biological roles. 

### 5.2. Cadmium Contributions to Metal Imbalance in Alzheimer’s Disease

Several trace metals have been implicated in AD pathogenesis and progression in addition to cadmium (reviewed in [81]). Zinc dysregulation induced by cadmium may be a risk factor for aggravating and advancing Alzheimer’s disease. Although research exists to probe the link between zinc and Alzheimer’s [82,83,84], little is known about the relationship between cadmium, zinc, and neuronal senescence. The imbalance of bioessential metal ions, particularly zinc, copper, and iron, has been observed in AD patients [85]. In a 2023 meta-analysis of 73 studies measuring levels of trace elements in AD patients, Li et al. reported alterations in the levels of copper in serum, iron in plasma, and zinc in hair [86]. Although Li et al. did not address cadmium levels, a 2017 meta-analysis of toxic metals in the circulation of AD patients reported increased cadmium levels as compared to controls [87]. Each aforementioned metal ion imbalance has consequences for neurodegeneration, and their complex, unique interplay may manifest as AD pathology. 

Imbalances of copper, iron, and zinc coupled with exogenous cadmium exposure appear to induce a cycle of exacerbated oxidative stress and promote toxic Aβ formation. As previously discussed, cadmium influx through zinc transports disrupts zinc homeostasis, intensifies oxidative stress, and downregulates the ZnT3 transporter [88]. A downregulation in ZnT3 is also observed in AD patients, indicating that zinc imbalance is closely linked to exogenous cadmium exposure [64,65]. Excess intracellular zinc not sequestered into vesicles by ZnT3 may then readily bind to Aβ monomers, promoting the production of toxic oligomers [60]. Recent research exploring the relationship between copper levels and AD has yielded mixed results. Some analyses report increases in copper levels of AD patients as compared to controls while other studies report no significant alterations in copper levels [86]. However, copper, specifically Cu(II), has been observed to bind to Aβ and contribute to plaques in the brain, which ultimately advances oxidative stress and neuroinflammation just as exogenous cadmium and zinc imbalances contribute to oxidative stress [89]. Iron also binds to Aβ monomers, establishing a structural change in these monomers that promotes toxic Aβ oligomer formation. Iron binds to Aβ via three histidine residues and one tyrosine residue in the N-terminal region of the Aβ monomer, which reduces the helical structure of Aβ and increases beta sheet content [90]. This structural alteration encourages the formation of toxic Aβ oligomers and aggravates neuroinflammation, which may then exacerbate iron imbalance and oxidative stress just as observed with respect to cadmium, zinc, and copper [91]. Cadmium, zinc, copper, and iron are all implicated in the progression of AD. The imbalance of each contributes to a perpetual cycle of oxidative stress and neuroinflammation that remains to be further researched and ultimately contributes to the production of toxic Aβ oligomers, a hallmark of AD pathology. 

## 6. Cadmium and the Blood–Brain Barrier

The BBB possesses a highly specific, tightly regulated architecture of polar epithelial cells that primarily rely on tight junction (TJ) formation for permeability control between cerebral vasculature and extracellular fluid of the nervous system. A collection of transmembrane and membrane-associated cytoplasmic proteins comprise TJs and act to control passive diffusion, restricting the entry of polar solutes into the CNS and anatomically separating CNS tissues from the bloodstream [20,92,93]. TJs are also regulated by pericytes, perivascular microglial cells, astrocytes, and neurons. The collective structural and modulatory elements of the BBB are known as the “neurovascular unit” (NVU), a term introduced in 2001 [94]. However, studies have reported that cadmium has caused disruptive alterations to the BBB that may underlie pathophysiologies in neurodegenerative disorders like AD, PD, and chronic traumatic encephalopathy, though the precise molecular mechanisms are not well understood [20,95].

Although cadmium easily crosses the immature BBB of young animals, it is typically restricted from crossing the adult BBB by strict TJ regulation [96]. However, accumulation in the adult brain does occur, particularly when cadmium is coupled with a vehicle that allows passage across the BBB such as ethanol [97,98], which commonly contains trace amounts of cadmium as an adulterant [5]. Interestingly, ethanol initiates a biochemical cascade to alter the permeability of the BBB in a similar fashion as cadmium, both beginning with an indirect upregulation of ROS that leads to a cellular stress response and culminates in a decrease of NVU protein expression [97,99,100]. 

Branca et al. reported that induction of oxidative stress occurred rapidly after treatment of a rat brain endothelial cell line (RBE4) with 10 µM cadmium chloride (CdCl_2_), with ROS production mediating an endoplasmic reticulum (ER) signaling pathway ultimately responsible for structural breakdown of tight junction and cytoskeletal BBB proteins [100]. Following exposure, ROS overproduction peaked at 5 min before returning to normal levels at 10 min and increasing again after two hours, indicating there may be dual short- and long-term oxidative stress responses following acute cadmium administration. Oxidative stress also activated an ER stress response, as evidenced by the authors’ investigation of GRP78, a well-studied chaperone protein indicative of ER stress, and found that CdCl_2_ exposure increased GRP78 expression three-fold as compared to controls. The stress response was followed by a significant upregulation of the apoptotic protein caspase-3 measured at 8 h post-exposure and abnormal immunocytochemical staining for three proteins that constitute tight junction and cytoskeletal architecture of the BBB: the zonula occludens-1 (ZO-1) protein, filamentous actin microfilament (F-actin), and vimentin. ZO-1 exhibited a loss of immunocytochemical staining, and stress fiber formation was visible for F-actin. Rupture and stretching of vimentin proteins were also observed. Disruption of these proteins via an oxidative stress-dependent ER stress response characterizes TJ disruption that ultimately results in secondary injury to the CNS similar to those observed in neurodegenerative diseases.

In 2021, Zhang et al. postulated an expanded mechanism by which cadmium disrupts BBB architecture [99]. Exposure of transgenic zebrafish embryos to CdCl_2_ (0, 10, 50, 100, or 500 µM) altered BBB morphology by disrupting endothelial cell–cell adhesion and inducing cerebral hemorrhage in a dose-dependent manner. Altered localization and function of BBB proteins ZO-1, vascular endothelial cadherin (VE-cadherin), and F-actin were due to a cadmium-induced oxidative stress cascade. However, this oxidative stress mediated the inhibition of protein tyrosine phosphatase (PTPase), an enzyme which regulates BBB integrity [101]. Oxidative stress has been observed to mediate tight junction damage caused by excessive protein tyrosine phosphorylation due to PTPase inhibition in human nasal epithelial cells [102]. In the zebrafish model, inhibition of PTPase also generated a rapid increase in the phosphorylation of VE-cadherin and ZO-1, initiating their displacement from typical BBB architecture. Inhibition of PTPase results in severe disruption of the BBB and proteolysis of occludin, explaining the increased BBB permeability exhibited by the embryos for 48 h post treatment and subsequent cerebral hemorrhage. Cadmium-induced destruction of the BBB architecture is a result of oxidant-induced cascades, and multiple downstream molecular mechanisms have been implicated in TJ and cytoskeletal disruption. Further research is needed to elucidate possible interactions between these mechanisms that contribute to the comprehensive BBB destruction observed after exogenous cadmium exposure.

We would be remiss not to acknowledge that both BBB perturbations and oxidative stress itself can exacerbate neuroinflammation, which is increasingly seen as a causal factor in myriad nervous system disorders, particularly as it relates to neurodegeneration. Though space prevents a full discussion of neuroinflammation and its deleterious effects across the lifespan, this topic has been excellently reviewed elsewhere [103,104,105,106,107]. However, one critical aspect of cadmium-associated neuroinflammation that warrants discussion is the activation of microglia by cadmium. Microglia exhibit macrophage-like functions in the brain, which include antigen presentation to T cells, general immune surveillance, and the secretion of pro-inflammatory cytokines such as TNF-α, IFN-γ, and IL-6 [108]. Cadmium can activate the excessively-damaging, pro-inflammatory functions of microglia by generating ROS and increasing the expression of NF-κB (a transcription factor involved in inflammatory responses) and upregulating caspase-3 (a protein involved in neuronal cell apoptosis) [109,110,111].

## 7. Cadmium’s Effects on Glycogen Metabolism

Glycogen metabolism in the brain is essential for significant central nervous system functions. Energy consumption in the brain is very high, with one study claiming that it accounts for 20–25% of the total body’s resting glucose consumption in adults. Developing brains likely requires an even greater percentage of this energy [112]. While glycogen is necessary as a source of readily available glucose to meet a neuron’s high energy demands, an overabundance of glycogen can lead to neurodegeneration, most commonly observed in glycogen storage diseases [113,114,115]. Thus, impairment of glycogen metabolism and storage is particularly harmful to the neurological function of an organism. Due to cadmium’s interference with cellular glycogen pathways, glycogen dysregulation represents a major avenue of its neurotoxicity.

Historically, hypotheses of cadmium’s neurotoxicity in regard to glycogen were based upon cadmium functionally impairing the glycogen phosphorylase (GP) enzyme, which catalyzes the first step of glycogen’s breakdown by facilitating the cleavage of glucose-1-phosphate (G1P) monosaccharides from the glycogen polysaccharide in glycogenolysis [116] (Figure 2). Glycogen phosphorylase exists as three isozymes in humans, of which brain glycogen phosphorylase (bGP) is the main glycogenolysis-facilitating GP isozyme of the central nervous system. Neurons express this bGP enzyme, and astrocytes express both bGP and the muscle GP isozyme [117]. Neurons contain measurable amounts of glycogen, but neighboring glial astrocytes are the primary glycogen-containing cells of the central nervous system [115]. Glycogen accumulation in neurons may actually be a marker of neurodegeneration [118]. Early research has suggested that cadmium’s neurotoxicity arose from inappropriately high accumulation of glycogen. While there is some merit and evidence supporting this original hypothesis, this idea may be inaccurate or present only a piece of the complete picture regarding cadmium’s glycogen-associated neurotoxicity. Recent evidence suggests that the depletion of glycogen reserves, not glycogen accumulation, is the main mechanism of cadmium’s glycogen-associated neurotoxicity.

### 7.1. Cadmium and Glycogen Phosphorylase Impairment: The Original Hypothesis

Cadmium has been hypothesized to impair the functionality of bGP, leading to the accumulation of glycogen in nervous system cells [116]. The basis for this hypothesis is derived from research demonstrating that thiol groups of cysteine residues in bGP are sensitive to metal ions such as cadmium [116,119]. As a result of glycogenolysis inhibition, an accumulation of glycogen in astrocytes may be a significant mechanism of neurological symptoms resulting from cadmium exposure.

Other inhibitors of glycogen phosphorylase such as CP-91149, CP-320626, and flavopiridol have been studied in the context of killing cancerous cells [120]. As a result of this enzymatic impairment, glycogenolysis was expectedly blocked, meaning that cells were unable to recycle glucose into the pentose phosphate pathways and were ultimately eliminated by apoptosis. Thus, the original hypothesis of cadmium-induced glycogen-mediated neurotoxicity is that cadmium interferes with the bGP structure at cysteine residues, decreasing enzymatic function, and blocking glycogenolysis, leading to inappropriately high glycogen storage, and eventually leading to neural cell death.

This hypothesis is supported by some experimental evidence. For example, in rats, cadmium acetate (CdAc_2_) intoxication at a concentration of 0.3 mg/kg of body weight has been found to disrupt the function of glycolytic enzymes, resulting in a 20% increase in glycogen accumulation when CdAc_2_ was subcutaneously injected twice weekly for three months [121]. Strikingly, the concentration of 0.3 mg/kg used in this study is well within the estimated range of the amount of cadmium in the typical adult human body (0.12 mg/kg to 0.5 mg/kg) [9]. Furthermore, cadmium exposure at a CdCl_2_ concentration of 0.49 mg/kg of pregnant Wistar rat body weight has been found to increase glycogen accumulation in rat placentae following daily injections until gestational age [122].

Information about the bGP enzyme itself also lends credence to the idea that cadmium could interfere with critical cysteine residues, ultimately disrupting enzyme function. Heavy metals such as cadmium have the ability to disrupt the function of enzymes reliant on cysteine due to heavy metal’s high affinity for sulfhydryl/thiol groups [17,90] (Figure 3(top)). The primary structure of human bGP reveals that there are 14 important cysteine residues of this enzyme, which is notable since 8% of human proteins do not have a single cysteine residue [92,123]. Moreover, the crystal structure of bGP had been solved with high resolution (2.5 Å and 3.4 Å) in 2016, which may help shine more light on this idea [113]. The structure highlights the diffuse nature of bGP’s cysteine residues and the relevance of the thiol group in multiple motifs of the native structure (Figure 3(bottom)). Thus, as a result of bGP’s multiple thiol groups from cysteines and cadmium’s ability to enter into the nervous system, bGP’s impairment is plausibly a result of cadmium interfering with its susceptible cysteine residues. Notably, these cysteine residues are not in the catalytic site nor the allosteric binding site of modulator AMP, though it is feasible that cadmium-induced modification of cysteine residues elsewhere in the protein negatively affects enzymatic activity. The likely mechanism by which this interference occurs is depicted in Figure 3, in which unoxidized cysteines important for the three-dimensional structure of bGP are instead coordinated to a cadmium cation.

Human clinical data also supports the theory regarding glycogen accumulation-mediated neurotoxicity. For instance, in the case of glycogen storage disease type IX, glycogen accumulation results from the inactivity of glycogen phosphorylase and has been observed to result in neurological symptoms such as ataxia and spasticity [124,125]. This rare genetic disease is characterized by a mutation affecting a phosphorylase enzyme responsible for activating GP. Though this disease does not show evidence of cadmium’s involvement in this process, it does show that GP downregulation is sufficient for neurotoxic effects. Other neurodegenerative conditions, such as Pompe disease, are associated with glycogen accumulation as well [126,127].

### 7.2. Evidence Contradicting the Original Theory Regarding Cadmium’s Functional Impairment of bGP 

While evidence does exist that supports the original hypothesis of cadmium’s glycogen-associated neurotoxicity, there also exists evidence in opposition to this theory. The Roelfzema study, which showed an increase in glycogen accumulation following cadmium exposure, paradoxically demonstrated a cadmium-induced increase in GP activity, indicating that while cadmium may lead to an increase in glycogen accumulation, it may do so in a mechanism not involving GP glycogen phosphorylase inhibition [122]. Furthermore, animal studies explored the effects of cadmium exposure on glycogen, and these studies have largely found that glycogen depletion, not accumulation, is likely the leading cause of cadmium’s glycogen-associated toxicity. For example, in climbing perch, cadmium exposure resulted in a significant reduction in glycogen levels in muscle and liver tissue, demonstrating that cadmium may affect the ability to store glycogen by inhibiting glycogen synthesis [128]. In freshwater bivalve mussel, exposure to CdCl_2_ (7.0 ppm and 12.0 ppm in water) increased glycogen degradation in its gastropod organs, thus increasing the rate of energy storage depletion in this species [87]. Furthermore, in rats, CdCl_2_ at a concentration of 2.6 and 5.2 mg/kg of body weight was found to reduce glycogen reserves in the liver, revealing that glycogen storage is impacted in mammalian species as well [129].

There is a notable lack of hypotheses regarding the mechanism by which cadmium induces glycogen depletion. The two general reasons by which cadmium would likely reduce glycogen reserves include (1) the impairment of glycogenesis by interfering with an enzyme involved in glycogen synthesis or (2) the excessive activation of glycogenolysis, which may be achieved by reducing the concentration of an inhibitor of glycogenolysis such as insulin. Interestingly, cadmium has been found to decrease insulin release, which may help explain the data that report reduced glycogen reserves as a result of excessive glycogenolysis in the absence of its insulin inhibitor [130]. Furthermore, glycogenolysis may be excessively activated as a result of cadmium interfering with PI3-kinase/Akt/mTOR signaling, which downregulates FOXO1, a transcription factor that stimulates glycogenolysis, and glycogen synthase kinase-3β, an enzyme that promotes glycogenolysis by inhibiting/phosphorylating glycogen synthase [131,132]. Akt/glycogen synthase kinase-3β signaling impacted by cadmium has also been linked to neuronal cell apoptosis [102]. Another theory simply states that cells in stressed conditions tend to need more energy to address the source of stress, thus increasing glycogenolysis to facilitate ATP-production from glucose stores [76,133]. The true mechanism is likely quite complex, encompassing the involvement of multiple enzymes and cellular processes.

### 7.3. Human Data Pointing towards Cadmium as a Glycogen-Disruptor

Human epidemiological data point towards cadmium exposure as a contributing factor to glycogen metabolism dysregulation. When glycogen metabolism is compromised by heavy metals exposure, including cadmium, there is a higher observed incidence of metabolic syndrome [134]. Furthermore, cadmium exposure also increases the risk of diabetes by affecting the glycogen/insulin pathways, which can lead to symptoms such as diabetic neuropathy [130,135,136].

Neurodegeneration can result from a prolonged period of glycogen metabolism dysregulation. This has been particularly observed in AD [137,138] and genetic diseases such as Lafora disease [139]. In AD pathology, glycogen synthase kinase-3β, which is upregulated as a result of cadmium toxicity, is considered to be a tau kinase, which contributes to the progression of the neurodegenerative disease [140]. Moreover, glycogen metabolism dysfunctions have been linked to schizophrenia, suggesting that cadmium could also lead to neuropsychiatric conditions such as schizophrenia as a result of its own metabolic dysfunction-inducing properties [141].

### 7.4. Next Steps for Resolving Cadmium’s Effects on Neuronal Glycogen

Overall, considering the evidence both for and against the original hypothesis that functional impairment of bGP leads to glycogen accumulation, it appears that cadmium may interfere with both glycogen synthesis and breakdown. The precise outcome on glycogen reserves, namely whether they are increased or decreased, is dependent on the particular species investigated and the tissue of interest. It is as yet unclear to what extent each of these mechanisms—glycogen depletion or glycogen overabundance—is most clinically and biologically relevant in the context of human cadmium neurotoxicity. However, it is clear from human clinical data that cadmium exposure can profoundly affect human health and disease in a manner that often involves aberrant glycogen metabolism. Future studies that must be performed to settle these discrepancies include neural tissue experiments that directly investigate glycogen reserves in glial cells following cadmium exposure, assays of the biological activity of all enzymes in both the glycogenesis and glycogenolysis pathways in response to cadmium exposure (i.e., not just glycogen phosphorylase), and examination of human histological samples following cadmium poisoning, when available.

## 8. Possible Protective Measures against Cadmium Neurotoxicity

Cadmium has long been recognized as a potent inducer of oxidative stress, disrupting the balance between ROS and antioxidants within the nervous system, thereby posing a significant threat to neural health. The preceding sections of this review have illuminated the intricate web of mechanisms through which cadmium exerts its neurotoxic effects, from mitochondrial dysfunction to cholinergic neuronal loss. However, a promising avenue of research has emerged, shedding light on the potential protective measures that antioxidants offer against cadmium-induced neurotoxicity.

Thiol-containing proteins, including glutathione (GSH), bovine serum albumin (BSA), and selenoprotein P, have also emerged as key players in mitigating cadmium-induced neurotoxicity. These proteins, known for their capacity to scavenge free radicals and bind to cadmium, may act as a protective shield against the harmful effects of cadmium on neural tissues [36]. By decreasing the availability of free cadmium to bind to critical thiol groups on mitochondrial proteins, these antioxidants help maintain mitochondrial function and prevent cadmium-induced permeability transition pore (PTP) opening, ultimately preserving neuronal integrity.

There is experimental evidence regarding specific antioxidants’ utility for mitigating cadmium-induced neurotoxicity. Branca et al. demonstrated that antioxidants have some combative power against cadmium, as application of the antioxidant α-tocopheryl acetate to rat brain endothelial cells exposed to cadmium prevented upregulation of GRP78, a marker of ER stress responsible for downstream damage to BBB architecture. The prevention of GRP78 upregulation by α-tocopheryl acetate supports that antioxidants have protective power against the oxidant-dependent ER stress response that ultimately disrupts BBB architecture and contributes to cadmium-induced neurotoxicity [100].

In a study utilizing Sprague–Dawley rats, cadmium exposure led to oxidative stress and autophagy within the testes. However, supplementation with the antioxidant quercetin demonstrated a protective ability to counteract Cd-induced testicular injury [142]. This finding highlights the potential of antioxidants to mitigate the adverse effects of cadmium on neural tissues. Furthermore, the neuroprotective potential of quercetin in the context of CdCl_2_-induced hippocampal neurotoxicity in male rats revealed that quercetin exerted a beneficial impact by enhancing memory function and mitigating hippocampal damage in CdCl_2_-treated rats. Quercetin increased the levels of antioxidants like glutathione (GSH) and manganese superoxide dismutase (MnSOD). Moreover, quercetin upregulated activity of SIRT1, a protein involved in cellular stress response, suppressed the activity of AChE, inhibited generation of ROS, and increased levels of brain-derived neurotrophic factor (BDNF), a protein crucial for neuronal survival and function [143].

Another noteworthy antioxidant, beta carotene, has shown promise in safeguarding neuronal health against the onslaught of cadmium-induced toxicity. In a comprehensive study on rats, cadmium exposure led to a significant increase in lipid peroxidation (LPO), indicative of oxidative damage within neural tissues. Cadmium exposure was also associated with elevated serum urea and blood urea nitrogen levels, indicative of renal dysfunction. Pre-treatment with beta carotene ameliorated the cadmium-induced increase in both LPO levels, serum urea, and blood urea nitrogen levels, underscoring its role in countering cadmium-induced oxidative stress and renal health [144].

In terms of cadmium clearance, Ethylenediaminetetraacetic acid (EDTA) has shown promising results. EDTA serves as a chelating agent extensively employed for the purpose of sequestering divalent and trivalent metal ions. EDTA binds to the metals through four carboxylates and two amine groups and forms especially strong bonds with Mn (II), Cu (II), Fe (III), and Co (III) [145]. Due to this property, EDTA is utilized as a medical treatment for the removal of lead and cadmium to mitigate metal toxicity [146]. Studies such as Waters et al., 2001, have shown beneficial results in EDTA chelation therapy where significantly higher urinary losses of cadmium were observed [147]. Whereas these studies were focused on the loss of cadmium from the body, recent studies by Fulgenzi et al., 2020 have dived into the neurotoxicity aspect of such. EDTA in the use of toxic metal chelation therapy were shown to have beneficial effects on neurodegenerative diseases, showing promising results for the future on protective measures against cadmium [148]. 

With recent evidence of Cd-induced neurotoxicity associated with increased ROS and mitochondrial-dependent ER stress, Mostafa et al., 2019 investigates the effect of rutin hydrate (RH), an antioxidant flavonoid well known as a neuroprotective substance [149]. Results showed that RH inhibits the mitochondrial permeability transition pore, enhances mitochondrial coupling, and inhibits mitochondrial cytochrome c release in the brain. Furthermore, RH inhibits mitochondrial Bax translocation, and, as previously discussed, Kim et al., 2013 demonstrated that Bax expression induces cell apoptosis, supporting that RH may be a potential protective measure against cadmium-induced neurotoxicity [47]. 

Cysteine has also been observed to reverse cadmium-induced blockade of skeletal neuromuscular neurotransmission [25]. Chick biventer cervicis nerve-muscle preparations exposed to 100 µM cadmium exhibited an 75% reduction in twitch heights within about 20 min of exposure, and application of 1 mM cysteine was able to fully reverse this blockade at the neuromuscular junction. The authors also conducted extracellular recordings of perineural waveforms at the motor nerve terminals of mouse diaphragm nerve-muscle preparations exposed to different concentrations of cadmium (10–100 µM) and reported that cadmium blocked the long-lasting positive deflection associated with calcium current. This cadmium-induced blockade was partially reversed by 300 µM cysteine and fully reversed by 1 mM cysteine. Cysteine may reverse the cadmium-induced block of calcium current by chelating cadmium, as cadmium has been reported to bind the thiol groups on cysteine residues of metallothioneins and the crucial antioxidant glutathione [150]. In a later review, Braga and Rowan stated that cysteine blocks all extracellular effects of cadmium, stressing the importance of developing cysteine therapies to mitigate cadmium neurotoxicity [25]. Combination therapies comprising exogenous chelating agents and antioxidants have been used to treat cadmium toxicity, but much remains to be explored in terms of best practices for the treatment and prevention of cadmium neurotoxicity [151]. Further studies should focus on the interactions between cadmium, cysteine, and other chelating agents in the nervous system to develop efficacious therapies that combat cadmium-induced neurotoxicity for both acute and chronic toxic exposures.

Zinc has also been suggested as a protective agent to counter cadmium neurotoxicity. Oral zinc supplementation is thought to prevent free radicals associated with cadmium-induced oxidative stress and alleviate cadmium-induced renal toxicity [151]. Oral supplementation has also been evidenced to slow the progression of neuronal senescence associated with cadmium toxicity and Alzheimer’s disease by reducing Aβ plaque formation in mouse models [152]. However, zinc’s protective functions with respect to neurotoxicity are much more complex and seem to be dose-dependent. Zinc supplementation at low doses protected rat hippocampal neurons from cadmium-induced disruption of neurotransmission but enhanced cadmium neurotoxicity at high doses [32]. More research is necessary to elucidate the intricate relationships between cadmium, zinc, Alzheimer’s disease, and oxidative stress in efforts to develop zinc-based therapies for cadmium neurotoxicity. 

## 9. Conclusions

Cadmium has several major routes of toxic insult to the nervous system (Figure 4). Cadmium first enters the body either via inhalation, whereby it can accumulate in the olfactory bulb, or, more commonly, via ingestion, eventually making its way to the bloodstream and the central nervous system via the BBB. Cadmium weakens the BBB, thereby increasing the opportunity for further cadmium entry. Cadmium can then enter cells and cellular compartments via calcium and zinc transporters. Intracellular cadmium can profoundly disrupt glycogen metabolism, alter neurotransmitter signaling, and disrupt mitochondrial function, which in turn increases the risk of neurodegenerative outcomes. 

There are several areas of cadmium neurotoxicity that merit further research. In all cases, more attention must be paid to the cadmium concentrations of these studies, as the current research represents a wide array of concentrations, some of which are orders of magnitude greater than the average human cadmium burden. First, more research is necessary to delineate the molecular relationship between cadmium, the PTP, and weakened ΔΨm. Further research must be conducted to elucidate cadmium’s complex interactions with both glycogenesis and glycogenolysis. Although current research suggests a tentative link between cadmium and neurodegenerative disease pathologies, particularly Alzheimer’s disease, future research should strive to extricate this link. In particular, further work regarding the role of cadmium in the development of neuronal senescence via its disruption of zinc homeostasis is directly relevant to neurodegenerative disease etiology. The identification of biomarkers common to both cadmium exposure and neurodegenerative disease may prove useful in development of targeted therapies for neurodegenerative disease amelioration and prevention. Research may also contribute to development of protective measures and therapies that alleviate neurotoxicity such as antioxidant and chelator combination therapy and oral zinc supplementation. 

As global industrialization continues to trend upward, the incidence of cadmium exposure is expected to increase. Physicians and public health officials must be aware of the neurotoxic effects of cadmium, which prove cumulative and potently toxic. 

Progress in understanding cadmium neurotoxicity will engender an understanding of and therapy development for both neurotoxicity itself and that of closely related, increasingly common neurodegenerative diseases. 

## Figures and Tables

**Figure 1 ijms-24-16558-f001:**
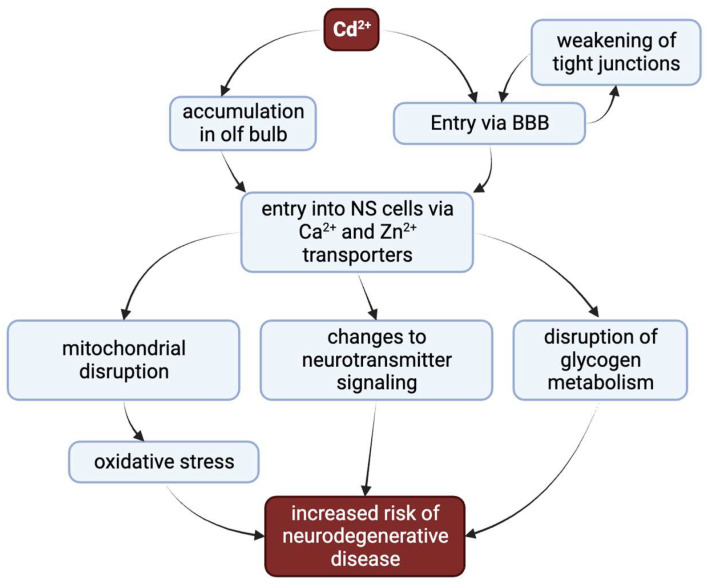
The diverse major pathways by which cadmium can increase risk for neurodegenerative disease. Cadmium accumulates in the olfactory bulb following inhalation. When either inhaled or ingested, cadmium passes into the bloodstream, which can decrease the integrity of the blood–brain barrier (BBB) via weakening of tight junctions. This allows cadmium to enter into nervous system tissue. Once within the nervous tissue, cadmium can efficiently pass the cellular membrane by co-opting transporters for other divalent cations. The primary mechanisms of neurotoxicity are disruption of glycogen metabolism, changes to neurotransmitter signaling, and mitochondrial disruption leading to oxidative stress. These perturbations together increase the risk for neurodegenerative disease.

**Figure 2 ijms-24-16558-f002:**
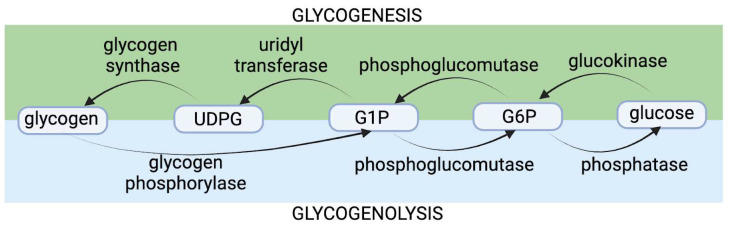
Glycogenolysis is the metabolic process by which cellular stores of glycogen are broken down into glucose. Glycogenesis is the opposite process, by which glucose is polymerized into glycogen. Both glycogenolysis and glycogenesis are mediated by critical enzymes, as shown.

**Figure 3 ijms-24-16558-f003:**
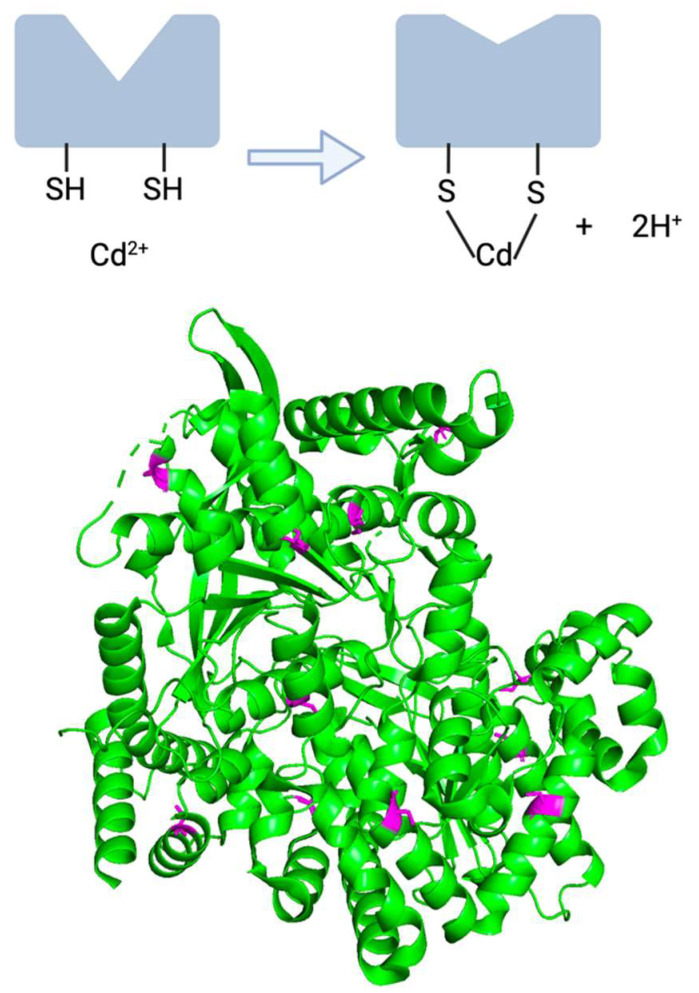
(**top**) Cadmium can interact at the sulfhydryl groups of cysteine residues, resulting in a change in enzyme structure. (**bottom**) bGP protein structure with cysteine residues highlighted (PDB entry: 5IKP).

**Figure 4 ijms-24-16558-f004:**
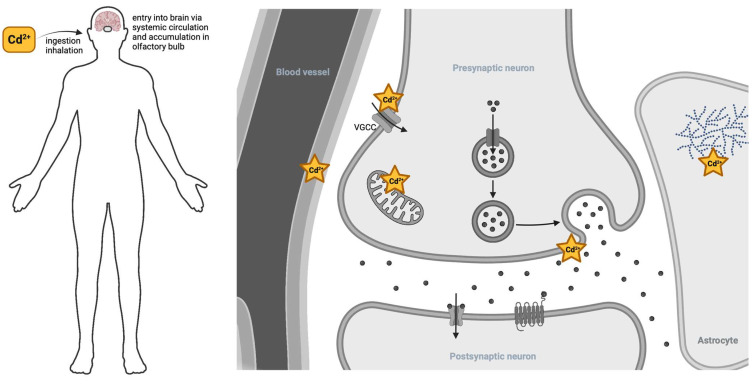
Cadmium’s entry to the nervous system (**left**). Cadmium gains entry to the nervous system through either inhalation or ingestion. Common sources are food products grown in soil with cadmium contamination. To a lesser extent, inhalation of cadmium fumes from industrial sources or cigarette smoke can enter through the lungs and olfactory bulb. Cadmium’s site of actions within the brain (**right**). Once in circulation, cadmium decreases the integrity of the blood–brain barrier, allowing further permeance into neural tissue. Cadmium gains entry into cells via cation-permeant integral membrane proteins, such as the voltage-gated calcium channel (VGCC). Once in the cytosol of astrocytes, cadmium disrupts glycogen utilization. Inside neurons, cadmium perturbs mitochondrial function and exocytosis of neurotransmitters via the vesicle cycle.

## Data Availability

Not applicable.

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
