# Peer review of "Mechanisms of Cadmium Neurotoxicity"

_ijms, 2023, doi:10.3390/ijms242316558_

Round 1

Reviewer 1 Report

Comments and Suggestions for Authors

The review is interesting and well-written. It presents some sections carefully curated. But some topics are only marginally covered or missing. Thus, the manuscript could be further improved before being published by addressing these weaknesses.

In the literature (for example PMID: 36362277), ample space is dedicated to autophagy in cadmium-induced tissue damage. The authors only mention it, when a short section on autophagy as acute and chronic toxicity of cadmium should also be dedicated in this case.

An important role of cadmium in the activation of microglia is reported in the literature. This topic has also been discussed in detail recently. The authors cite only the reference (PMID: 36760476), while they could dedicate at least one section, given the importance of these cells in neurotoxicity processes (PMID: 37371389).

The authors have only marginally addressed the topic of metal imbalance in Alzheimer's. This is a relevant neurodegenerative disease that deserves a dedicated section.  Even if their study is basic research, it is important to frame the topic in the frameshift of existing literature including clinical studies. clinical studies are relevant since they connect basic research to therapeutics. Clinical studies or better meta-analysis studies, which represent a consensus of the studies published on the recent meta-analyses addressed cadmium and other metals involvement in AD (PMID: 29439342;PMID:34209820; PMID: 36464120). As a matter of fact, the authors discussed cadmium as if it were a metal in itself and dedicated little space to the interaction of this metal with others that are relevant in the most widespread and most studied neurodegenerative disease, such as Alzheimer's. a short paragraph on how the most representative metals (Cadmium, Cu, Zn, Fe) for AD are interconnected could be added

A major issue is figures of mechanisms of Cadmium toxicity that are missing. Authors should present at least only one figure representing the main mechanisms of cadmium toxicity at cell levels. Also, the Figure 1 legend is insufficient.

 The conclusions are too long. They should just be short paragraphs with a take-home message for the reader for each of the sections presented, but instead, they introduce new concepts. This happens between riches 698-711 for example.

Authors can move topics that are not conclusions to other paragraphs and shorten the conclusions to the essentiality 

Reviewer 2 Report

Comments and Suggestions for Authors

Authors need to include more figures and tables.

Comments on the Quality of English Language

English is fine.

Round 2

Reviewer 1 Report

Comments and Suggestions for Authors

The authors made many improvements to the manuscript. It is now suitable for publication

Author Response

Thank you for your valuable feedback. We appreciate your guidance which has improved this manuscript such that it is now suitable for publication.